# Identification of Small Molecule Inhibitors against Mycobacteria in Activated Macrophages

**DOI:** 10.3390/molecules27185824

**Published:** 2022-09-08

**Authors:** Rebecca Vande Voorde, Elizaveta Dzalamidze, Dylan Nelson, Lia Danelishvili

**Affiliations:** 1Department of Biomedical Sciences, Carlson College of Veterinary Medicine, Oregon State University, Corvallis, OR 97331, USA; 2Department of Pharmaceutical Sciences, College of Pharmacy, Oregon State University, Corvallis, OR 97331, USA; 3Department of Microbiology, College of Science, Oregon State University, Corvallis, OR 97331, USA

**Keywords:** *Mycobacterium avium*, *Mycobacterium abscessus*, *Mycobacterium tuberculosis*, macrophage, high-throughput screen, small molecules, intracellular, non-replicating, drug resistance, virulence factors

## Abstract

Mycobacterial pathogens are intrinsically resistant to many available antibiotics, making treatment extremely challenging, especially in immunocompromised individuals and patients with underlying and chronic lung conditions. Even with lengthy therapy and the use of a combination of antibiotics, clinical success for non-tuberculous mycobacteria (NTM) is achieved in fewer than half of the cases. The need for novel antibiotics that are effective against NTM is urgent. To identify such new compounds, a whole cell high-throughput screen (HTS) was performed in this study. Compounds from the Chembridge DIVERSet library were tested for their ability to inhibit intracellular survival of *M. avium* subsp. *hominissuis* (MAH) expressing dtTomato protein, using fluorescence as a readout. Fifty-eight compounds were identified to significantly inhibit fluorescent readings of MAH. In subsequent assays, it was found that treatment of MAH-infected THP-1 macrophages with 27 of 58 hit compounds led to a significant reduction in intracellular viable bacteria, while 19 compounds decreased *M. abscessus* subsp. *abscessus* (Mab) survival rates within phagocytic cells. In addition, the hit compounds were tested in *M. tuberculosis* H37Ra (Mtb) and 14 compounds were found to exhibit activity in activated THP-1 cells. While the majority of compounds displayed inhibitory activity against both replicating (extracellular) and non-replicating (intracellular) forms of bacteria, a set of compounds appeared to be effective exclusively against intracellular bacteria. The efficacy of these compounds was examined in combination with current antibiotics and survival of both NTM and Mtb were evaluated within phagocytic cells. In time-kill dynamic studies, it was found that co-treatment promoted increased bacterial clearance when compared with the antibiotic or compound group alone. This study describes promising anti-NTM and anti-Mtb compounds with potential novel mechanisms of action that target intracellular bacteria in activated macrophages.

## 1. Introduction

Nontuberculous mycobacteria (NTM) are a large, ubiquitous group of bacteria of the genus Mycobacteria capable of causing human disease and death [1,2,3]. Although mainly affecting immunocompromised individuals, NTM are increasingly found to be the cause of pulmonary, gastrointestinal, and skin infections in immunocompetent adults and children [2,4]. Unfortunately, treatments for these pathogens are complicated by acquired resistance to many conventional antibiotics and by their innate ability to render them ineffective [5]. Such innate mechanisms include the ability of bacteria to create an impermeable cell wall as well as the presence of many efflux pumps and several drug-modifying enzymes [6,7,8,9]. In addition, host environmental stresses influence mycobacterial physiology and metabolism and alter bacterial cell surface, further promoting the persistence phenotypes and tolerance to many antibiotics [10,11,12,13].

*M. abscessus* (Mab) is the most common species of NTM pulmonary infections in patients with chronic lung diseases such as cystic fibrosis (CF), bronchiectasis, previous history of tuberculosis, or chronic obstructive pulmonary diseases (COPD) [1,5]. Treatment of Mab generally requires the long-term administration of multiple antibiotics. Most often two macrolides, such as clarithromycin and azithromycin, along with an aminoglycoside, commonly amikacin, and the cephalosporin cefoxitin are used for the initial phase of treatment [14]. After two weeks to several months of the initial phase, nebulized amikacin and a combination of oral macrolides are given until sputum samples become negative for 12 months [15]. Despite this intensive treatment regimen, the infection is successfully resolved in only 25% to 40% of cases [16]. The high treatment failure rates in clinics highlight the inability of currently available antimicrobials to effectively clear Mab infections in the host.

The *Mycobacterium avium* complex (MAC) is another species of NTM that causes serious pulmonary infections in persons with respiratory comorbidities [3,17]. The initial treatment for MAC infections is a multidrug regimen similar to that used for Mab infections. However, in cases of macrolide-resistant MAC or patients with severe bronchiectasis, oral administration of azithromycin, rifampicin, and ethambutol is combined with amikacin or streptomycin for up to three months [15]. As seen for Mab, even with this intensive antibiotic regimen therapy success rates for MAC are very low. Treatment failure occurs in 20% to 40% of patients and up to 50% of individuals have recurring disease [5].

It is apparent that the treatment regimens with antibiotics currently used in the battle against NTM infections are suboptimal, and inconsistencies between in vitro and in vivo efficacy emphasize the work needed toward the development of innovative therapeutic strategies that can aid, improve, or amplify the clinical efficacy of current antibiotics toward (a) effective elimination of intracellular NTM pathogens and (b) shortening the lengthy therapy regimens to reduce adverse effects and the development of acquired drug-resistance.

Several high-throughput compound screening assays have been performed utilizing in vitro mycobacteria grown in various culture media and conditions [18,19,20,21,22]. While many classes of antibacterial compounds have been identified in this way, the approach can miss compounds that may target important bacterial or host factors exclusively expressed in different physiologic stages of intracellular bacilli infection. For example, recent proteome research examining *M. avium* subsp. *hominissuis* (MAH) response to antibiotics demonstrates a wide range of variability in the protein profile and limited metabolic pathways that are activated under diverse environmental conditions [12]. Similar adaptive changes have been reported in Mab metabolism during growth in biologically relevant host conditions and during treatment of antibiotics, giving some insights on how bacteria may tolerate to antibiotics [10]. Therefore, cell-based HTS screens are an attractive approach for the discovery of compounds that can target bacterial or host cellular factors exploited by the pathogen for entry, replication, or intracellular survival and, thereby, affect mycobacterial pathogenicity and virulence.

Because phagocytic cells such as macrophages represent physiological conditions mimicking disease and contribute to the process of NTM eradication, in this study, we initiated ex vivo high-throughput screening to evaluate small molecule compound activity against fluorescently labeled *M. avium* subsp. *hominissuis* during the infection of THP-1 macrophages. Several hit compounds were identified and tested against intracellular Mab and *M. tuberculosis* as well. While the majority of compounds showed activity against the bacteria of intracellular and extracellular phenotypes, some selected compounds exhibited activity only in the intracellular environment. The combination of these compounds with current antibiotics led to increased efficacy of current antibiotics and higher bacterial clearance when compared to antibiotic treatment alone.

## 2. Materials and Methods

Chemicals and compound libraries. The ChemBridge DIVERSet small molecule compound library was obtained from the Oregon State University/College of Pharmacy High-Throughput Screening Services Laboratory in a single dose of 10 μM and a final DMSO concentration of 1%. Amikacin was purchased from Research Products International, (Mt. Prospect, IL, USA), and clarithromycin from Tokyo Chemical Industry, (JAPAN). Phorbol 12-myristate 13-acetate (PMA) and resazurin were obtained from Sigma (St. Louis, MO, USA). Metal mix media has been previously formulated and described [23,24,25].

Bacterial strains and cell lines. *Mycobacterium avium* subspecies *hominissuis* 104 (MAH104) was isolated from the blood of an AIDS patient [26]. MAH104 was mouse-passaged to maintain virulence and was used for fewer than 10 passages on 7H10 Middlebrook agar plates. *Mycobacterium abscessus* 19977 (Mab19977) and *Mycobacterium tuberculosis* H37Ra (MtbH37Ra) were purchased from the American Type Culture Collection (ATCC). Clinical isolates of MAH strains 0133, MAH-B, and MAH-C and Mab DNA01627, NR49093 strain DJO44274, and NR44273 strain 4529 were obtained in collaboration with the Cystic Fibrosis Research and Development Program at National Jewish Health in Denver, CO, USA. Mycobacteria were grown on 7H10 Middlebrook agar or 7H9 Middlebrook broth (Difco Laboratories, Detroit, MI, USA) containing 10% oleic acid, albumin, dextrose, and catalase (OADC, Hardy Diagnostics, Santa Maria, CA, USA) and glycerol. MAH expressing the tdTomato protein (MAH-T) was constructed in this laboratory by electroporation of pJDC60 plasmid into MAH104 using standard protocols [27]. MAH-T was grown as described for MAH104 except with the addition of 400 μg/mL kanamycin to maintain the plasmid. To make suspensions of these strains, log phase cultures grown on a 7H10 agar plate were suspended in 10 mL Hanks’ Balanced Salt Solution (HBSS; VWR, Visalia, CA, USA). Cells were passed through a 22-gauge syringe 10 times to disperse clumping and were allowed to settle for 15 min. The top 9 mL was removed to a new tube. The previous two steps were repeated, and the top 6 mL was removed to a fresh tube. The O.D. 600 was read and cells were diluted in RPMI-1640 medium to the appropriate concentration (1.0 O.D._600_~3 × 10^8^ cells/mL).

Human-derived monocytic THP-1 cells (ATCC TIB-202) were grown in RPMI-1640 medium (Cellgro, Manassas, VA, USA) supplemented with 10% heat-inactivated fetal bovine serum, 2 mM L-glutamine, and 25 mM HEPES (termed as RP10) and maintained at 37 °C with 5% CO_2_. Cells were grown at a concentration less than 1 × 10^6^/mL and kept for fewer than 10 passages. To differentiate THP-1 cells into macrophages, cells were plated at 1 × 10^5^/well in a clear, flat-bottom 96-well plate in RP10 and treated overnight with 50 ng/mL PMA. The following day, media was removed and 100 μL fresh RP10 was added. Cells were incubated for an additional two days at 37 °C/CO_2_ before infection.

Optimization of the high-throughput screening (HTS) assay. To optimize the HTS screen, different bacterial multiplicity of infections (MOIs) and infection durations were tested in THP-1 cells. Differentiated THP-1 cells were seeded at 10^5^ in 100 μL RP10 in a Corning clear flat-bottom 96-well plate. Plates were incubated overnight at 37 °C/CO_2_. The following day media was removed, and fresh media was added. Forty-eight hours later, two-fold dilutions of a single cell suspension of MAH-T were made in RP10 and added to give a final MOI of 5, 10, 20, or 40 bacteria to 1 cell. After 2 h, cells were washed with HBSS, and 200 μg/mL amikacin was added for an additional 2 h to kill extracellular bacteria. Next, wells were replenished with fresh RRP10 containing 400 μg/mL kanamycin. As a control, 70 μg/mL clarithromycin was added to some wells. Cells were incubated up to 10 days and red fluorescence was measured daily using 530 nm/590 nm excitation/emission filters on a Tecan 200 fluorimeter.

The HTS primary assay. The PMA-stimulated THP-1 cells were seeded in 96-well plates as described above and infected at an MOI of 40 bacteria to 1 cell. The inoculum of MAH-T was prepared in RP10, and cells were infected for 2 h. Next, cells were washed with HBSS, treated with 200 μg/mL amikacin in RP10 for 2 h to remove extracellular bacteria, and replenished with RP10 media containing 400 μg/mL kanamycin together with tested compounds from the ChemBridge small molecule library at the final concentration of 10 μM excluding the last column. RP10 containing 400 µg/mL kanamycin and 70 μg/mL clarithromycin was added to the last column as a control for 100% inhibition. DMSO was also included as an additional control for measuring any effects on intracellular bacterial growth. Plates were incubated at 37 °C/CO_2_ and fluorescence at 530/590 nm excitation/emission was measured on a Tecan plate reader on days 5, 6, and 7 of post-infection. Fluorescence data of individual wells was analyzed as a percentage of the DMSO control using the formula: % inhibition = 100 × [1 − (X − MIN)/(MAX − MIN)] where MIN is the average of the clarithromycin-treated wells and MAX is the average of the DMSO-treated wells. Compounds that inhibited bacterial fluorescence for more than 50% on days 5, 6, and 7 were chosen as potential hits. The hit compounds were selected from the library and were retested in the secondary assay in three technical replicates as described above. The compounds that produced similar results in primary and secondary screens were reordered from the ChemBridge corporation for further studies.

Cytotoxicity assay. To establish the cytotoxic concentration of compounds causing 50% or more death of viable host cells, the hit compounds were tested in THP-1 differentiated macrophages in a concentration range of 10 to 0.01 μM that was made in DMSO by half-log serial dilutions. Four microliters of compound dilutions were added to 196 μL RP10 and dispensed on cell monolayers in each well of 96-well plates. Cyclosporin A at 50 μg/mL was used as a positive control for cytotoxicity and 1% DMSO as a negative control. Plates were incubated at 37 °C/CO_2_ and 72 h later assessed with a resazurin colorimetric assay [28]. Ten microliters of resazurin (50 μg/mL in PBS) were added to each well and incubated until an appropriate color change occurred (~3–5 h). The fluorescence was read at 530/590 nm in a Tecan plate reader, and the percent of inhibition was determined using the formula described above. In addition, cell monolayers were visually evaluated for cytotoxicity using an inverted microscope. Toxicity was determined by the least concentration of compound that did not cause a color change in resazurin, and/or the least concentration in which no visible effects on the cell monolayer were observed.

MIC50. To determine the minimum inhibition concentration for inhibiting 50% of the pathogen, bacterial suspensions of MAH104, MAB19977, and MtbH37Ra, or clinical isolates were made in 7H9 media containing OADC, 10% glycerol, and 0.1% Tween. The inoculums were adjusted at OD_600_ to 0.5 and diluted to 0.03–0.05 in 7H9 media prior to the compound addition. Compounds in a concentration range of 10 to 0.01 μM were made in DMSO by half-log serial dilutions and added to corresponding wells. Clarithromycin (70 μg/mL) and 1% DMSO served as controls. The experiment was performed in duplicate plates. Plates were incubated statically for 72 h for MAB19977, 96 h for MAH104, or 144 h for MtbH37Ra. Resazurin was added to plates at a final concentration of 5 μg/well and incubated at 37 °C until an appropriate color change occurred. The fluorescence was read at 530 nm/590 nm on a Tecan plate reader, and the percent of inhibition was determined using the formula described above.

Quantification of intracellular mycobacteria in compound alone or compound-antibiotic combination treatment of macrophages. To establish intracellular bacterial growth within macrophages with and without compound treatment, THP-1 macrophage monolayers were infected with either MAH104, Mab19977, or MtbH37Ra. Infections were carried out for 2 h with an MOI of 5 for MAH104 and MOI of 1 for Mab19977 and MtbH37Ra. After infection, cells were treated with 200 μg/mL amikacin for 2 h to remove any extracellular bacteria. Wells were replenished with fresh RP10 containing tested compounds alone or in combination with 4 μg/mL amikacin for MAH104 and Mab19977, and 0.2 μg/mL isoniazid (INH) for MtbH37Ra. Plates were incubated at 37 °C/CO_2_ up to six days and the number of surviving bacteria was enumerated by lysing cells with 0.1% Triton X-100 at the time points indicated in the figure legends. Serial dilutions were made in HBSS and plated on 7H10 agar plates for 7–14 days. Data were calculated as a percentage of surviving bacteria in the DMSO alone control. The statistical analysis and graphical outputs were made in the GraphPad Prism (version 9.0) with one-way ANOVA multiple comparisons between DMSO control and experimental compound treatment groups.

Antimicrobial activity of compounds against mycobacteria in metal mix media. Single-cell suspensions of mid-log phase grown MAH104, Mab19977, or MtbH37Ra were diluted to O.D. 600 = 0.05 in the 7H9 growth or metal mix (MX) media [23]. Compounds were tested in 96-well plate format at 100 μM concentration, while 1% DMSO was used as a positive control for bacterial growth and 70 μg/mL clarithromycin or INH (Mtb) as a negative control for 100% bacterial growth inhibition. Plates were incubated statically at 37 °C and after 72 h (for MAH and Mab) or 120 h (for Mtb), the O.D. 600 nm readings were recorded. In addition, to analyze the viable bacteria, resazurin at a final concentration of 5 μg was added to each well. After appropriate color change, plates were visually examined and read at 530/590 nm excitation/emission filters on a fluorimeter (Tecan).

## 3. Results

A fluorescence-based HTS identifies compounds active against intracellular mycobacteria. A cell-based high-throughput screen was initiated using infected THP-1 macrophages with MAH-T expressing tdTomato protein and viable bacterial growth within the host cells was measured through fluorescence readings. The HTS screening triage summarizing the primary and counter assays is detailed in Figure 1A. The primary HTS assay was optimized in a 96-well format using different MOIs and infection time points to determine the conditions that led to the greatest difference in fluorescence between DMSO- and clarithromycin-treated wells. Infection of differentiated THP-1 cells with MAH-T at an MOI of 40 at 5, 6, and 7 days of infection provided the most consistent difference in fluorescence between the positive (DMSO) and negative (CLA) controls (Figure 1B,C). The Z’-score was determined to be 0.55, which is considered a good quality assay [29].

In the primary screen, 40,560 compounds from the ChemBridge DIVERSet library obtained from OSU/College of Pharmacy High-Throughput Screening Services Laboratory were added to MAH-T-infected macrophage monolayers at a final concentration of 10 μM in 0.1% DMSO. Eighty compounds per 96-well plate were screened and the percentage of intracellular bacterial inhibition, compared to DMSO alone control wells, was calculated on days 5, 6, and 7 of MAH-T infection. Those compounds showing more than 50% inhibition in red fluorescence on all three days were considered potential hits. Out of 40,560 compounds tested, 731 compounds exhibited a more than 50% decrease in the fluorescence of infected THP-1 cells (hit rate of 1.8%). In repeat assays performed in three technical replicates, the activity of 58 compounds (out of 731) was confirmed to significantly decrease MAH-T fluorescence in infected macrophages (repeat rate of 8%). These 58 compounds were purchased from the ChemBridge company (San Diego, CA, USA) and further evaluated for their activity against mycobacteria based on quantification of intracellular CFUs and for MIC50 assays as described below.

Quantification of intracellular bacteria confirms the antimicrobial activity of hit compounds in macrophages. The summary of compounds that significantly reduced intracellular mycobacterial loads (either NTM or Mtb) in infected macrophages and/or exhibited activity in vitro are listed in Table 1. These compounds belong to 17 different classes of chemical groups: thiazole- and pyridine-hydrazides, triazole- and thiophene-carboxamides, pyrimidine, phenyl-urea and thiourea, triazine diamine, benzamide, and others. Some of these compound classes have already been shown to be active in *M. tuberculosis.* The primary screen also identified isoniazid and ciprofloxacin analogs (compounds #**4** and #**6**, respectively), validating our HTS screen.

The cytotoxicity of 58 compounds was tested in differentiated THP-1 cells in the concentration range of 0.1 to 100 μM using a resazurin colorimetric assay together with the visual examination of macrophage monolayers. Bacterial survival assays were performed at the highest concentration of compounds that were nontoxic to THP-1 cells (Table 2).

Of the 58 tested compounds, 27 showed a statistically significant reduction in intracellular MAH104 growth recorded as colony-forming units (CFUs) over 6 days (Table 2 and Figure 2A). While 19 out of 58 compounds decreased viable Mab19977 loads over 3 days of infection, 14 active compounds inhibited intracellular MtbH37Ra survival at day 6 post-infection of THP-1 cells (Table 2 and Figure 2B,C). Five of these active compounds were found to be common for tuberculosis and non-tuberculous mycobacteria, and 15 compounds decreased the survival of both NTM species but not Mtb. In addition, 2 compounds were found to have activity exclusively in MAH and 3 compounds only in Mab. Furthermore, 7 compounds exhibited activity in MAH104 and MtbH37Rv, but no activity in Mab (Table 2 and Figure 2). We should highlight that for some compounds with intracellular and/or extracellular potency, the combination together with bacterial infection led to increased toxicity to THP-1 cells (at nontoxic concentration). These compounds are marked with asterisks and were included in Table 2 because they display in vitro activity and are members of clusters.

In vitro antimicrobial activity of hit compounds against replicating mycobacteria. To determine whether the identified 58 hit compounds also displayed activity against bacteria in vitro, half-log dilutions of compounds were made in 1% DMSO and added to the mid-log-phase grown cultures of MAH104, Mab19977, or MtbH37Ra in 7H9 broth. Viable bacteria were assessed with OD_600_ and by oxidation rates of resazurin substrate on days 4, 3, and 6, respectively. The in vitro active compounds and established MIC_50_ concentrations are listed in Table 2. Our results demonstrate that 18 compounds exhibited more than 50% inhibitory activity against MAH104 in 7H9 growth media, including ciprofloxacin and isoniazid analogs. Eight compounds showed activity against Mab19977 and 15 compounds against MtbH37Ra. While compounds and MIC_50_ concentrations varied for both NTM organisms and Mtb, three common compounds exhibited in vitro activity in all tested mycobacterial species (Table 2).

We also evaluated hit compounds against clinical isolates of MAH and Mab drug-resistant strains of cystic fibrosis patients. MAH strains of 0133, MAH-B, and MAH-C are resistant to amikacin (this study), and Mab isolates of DNA01627, NR49093 strain DJO44274, and NR44273 strain 4529 have been previously shown to be resistant to amikacin and clarithromycin [6]. The MIC assay was performed as described in the materials and methods section, and the results demonstrate that while the majority of compounds have similar MIC concentrations, some are even more susceptible to selected compounds (Table 3). Interestingly, compounds **5**, **10**, **31**, **44**, and **47** with activity exclusively in intracellular bacteria that were identified in MAH104 and Mab19977, did not display activity across clinical isolates as well (data not shown).

Cotreatment with hit compounds and antibiotics improves bacterial clearance within macrophages. We carried out bacterial survival studies with frontline antibiotics (AMK for NTMs; INH or RIF for Mtb) and compounds that showed activity exclusively in infected cells but not in vitro. To determine whether these compounds demonstrated any synergy or antagonism with antibiotics, infected THP-1 cells were treated with either compound or antibiotic alone, or compound with the antibiotic combination. In vitro MIC concentrations of antibiotics were used in the experiment to determine if combination treatment with compounds could lead to improved antibiotic efficacy and, therefore, better clearance of bacteria when compared with the antibiotic alone control group. As shown in Figure 3A, the combination of two drugs led to substantially less intracellular MAH growth at later time points. Similar results were obtained when the combination treatment of compounds with and without antibiotics was tested against Mab and Mtb (Figure 3B,C).

Evaluation of antimicrobial activity of compounds using an in vitro model of a phagosome. The compounds that were active only against intracellular bacteria, and not in vitro, were tested in MX media. MX mimics the metal concentrations and pH of the mycobacterial vacuole [25], and has been shown to stimulate gene expression as well as secretion of several potential virulence factors [23,24]. Most importantly, it stimulates the development of the persistent state in mycobacteria and promotes tolerance to currently available antibiotics [30].

Interestingly, three compounds, **5**, **10**, and **47**, which did not show any activity in 7H9 growth media were highly bactericidal for MAH104 in MX media, and compounds **10** and **47** had similar potency for Mtb in MX but not in 7H9 broth (Figure 4A,C), demonstrating that these compounds retain activity in conditions similar to those found within the phagosome vacuole. Compound **5** showed significant inhibition of Mab19977 in MX media while compounds **10** and **47** caused a slight decrease in bacterial numbers (Figure 4B). The differences in the effects of active compounds on MAH in metal mix compared to Mab are unclear. However, we should emphasize that MX was mainly designed with two formulations to mimic the phagosome environment of MAH and Mtb, but not Mab [25].

## 4. Discussion

Mycobacterial pathogens are predominantly intracellular organisms, causing morbidity and mortality. Mycobacteria can invade and proliferate within a variety of mammalian cells, including mucosal epithelial cells and phagocytes [31,32,33]. However, both Mtb and NTM organisms preferentially infect macrophages where they establish a persistence niche within phagosome vacuoles for prolonged survival and dissemination in the host [34,35]. The intracellular environment of the phagosome where bacteria reside significantly differs from the extracellular condition of the growth culture [7,8]. The transcription, as well as proteome profiles of persistent mycobacteria, demonstrates the temporal regulation of a subset of genes and synthesis of distinct proteins within the host in response to environmental cues that collectively enable the intracellular bacteria to become phenotypically resistant and avoid killing by antimicrobials [10,11,12,36]. Therefore, identifying compounds that can exhibit antimicrobial activity within the cellular environment of the host is an important attribute in mycobacterial drug discovery.

Finding compounds that target essential bacterial factors expressed in various conditions and stages of host infection is challenging. To address this, cell-based assays have emerged as a critical tool in the search for intracellular inhibitors. Several high-throughput screening campaigns designed to identify inhibitors of mycobacterium species have been conducted. Many of these were performed in vitro using bacteria grown in various types of laboratory media and different environmental conditions [18,20]. Others targeted specific proteins or pathways of mycobacteria [19,21,22]. Only a few HTS campaigns were designed to identify inhibitors of intracellular mycobacteria [22,37,38].

The use of infected macrophages has the advantage of identifying intracellular bacterial-specific targets that may serve as virulence factors as well. Furthermore, infected human cell lines offer the additional benefit of discovering compounds with activity against cellular targets that could be utilized by mycobacteria for its survival and pathogenicity in the host. Therefore, in this study, a cell-based high-throughput screen was initiated using THP-1 human macrophages. *M. avium* expressing the fluorescent tdTomato protein was used to easily measure bacterial replication in host cells through fluorescence readings. Compounds that decreased the red fluorescence of infected THP cells were considered to be potential inhibitors of MAH infection. While this screen would select compounds that naturally inhibit fluorescence, these were ruled out in subsequent secondary assays by retesting compounds for their ability to decrease mycobacterial CFUs in macrophages. By testing activities for both Mtb and NTM organisms, we aimed to obtain compounds with broad anti-mycobacterial spectrum potency and possibly with activity against common bacterial targets or host pathways potentially contributing to the persistence of these pathogens.

In this study, the chemically diverse library was selected from the larger screening collection of the ChemBridge Corporation as it was shown to contain several active inhibitors of Mtb in vitro and, therefore, it may also contain compounds active in vivo [20]. The primary HTS screen of over 40,000 compounds in MAH-T infected human macrophages identified 731 compounds that cause a more than 50% decrease in bacterial fluorescence. The activity of 58 compounds was confirmed in repeat fluorometric assays, and selected compounds listed in Table 1 were purchased from ChemBridge and evaluated in subsequent assays. A total of 27 of 58 tested compounds showed a significant reduction in intracellular MAH104 growth recorded as CFUs, while 19 and 14 compounds were found to be active against Mab19977 and MtbH37Ra, respectively. Overall, 36 active compounds were confirmed to significantly reduce intracellular loads of either MAH, Mab, or Mtb within THP-1 macrophages with five compounds overlapping between mycobacterial species. Based on the primary skeletal moieties, these compounds were grouped into 17 classes of thiazole- and pyridine-hydrazide, triazole- and thiophene-carboxamide, pyrimidine, phenyl-urea and thiourea, triazine diamine, benzamide, and others listed in Table 1.

To define whether the 36 compounds identified also displayed antibacterial activity in actively replicating mycobacteria in growth culture, they were tested in vitro in 7H9 broth. MIC assays established that 18 compounds have in vitro bactericidal activity against MAH, 8 against Mab, and 15 in Mtb at various concentrations listed in Table 2. Compounds **5**, **10**, **31**, **44**, and **47** did not display activity against MAH or Mab in vitro, nor did compounds **10** and **47** against Mtb. These compounds were also examined in MX media in vitro. MX mimics the metal concentrations and pH of the mycobacterial phagosome [25] and was shown to stimulate the persistence phenotype in mycobacteria and tolerance to antibiotics [30]. Compounds **5**, **10**, **31**, **44**, and **47** demonstrated bactericidal activity against MAH in MX, while only three compounds (**5**, **10**, and **47**) inhibited Mab growth in MX. In addition, compounds 10 and 47 displayed inhibitory activity against Mtb in MX media.

Compounds **5** and **10** belong to the triazine diamine cluster. Previous work identified a series of novel diamino triazines that were found to be potential inhibitors of Mtb dihydrofolate reductase enzyme, which protects cells against growth inhibition by isoniazid by sequestering the drug [39]. Furthermore, the literature reveals that antitubercular triazines stimulate the release of intrabacterial NO and desnitro metabolites as a mechanism of action along with inhibition of *Inh*A encoding a target for isoniazid and ethionamide and the FAS-II pathway in Mtb [40]. Compound **31** belongs to the triazole-carboxamide cluster and compound **44** to phenyl-urea. The triazole-carboxamides have been demonstrated to suppress bacterial SOS responses through the inhibition of the dual-function repressor/protease LexA and block a mechanism enabling *Escherichia coli* and *Pseudomonas aeruginosa* to adapt and resist antimicrobial treatment [41]. It is reported that the adamantyl-phenyl-urea compounds target MmpL3 protein that is involved in the secretion of trehalose mono-mycolate in Mtb [42]. Recently, through whole cell phenotypic HTS, a phenyl-urea compound was identified as one of the lead inhibitors of MmpL3 with high potency against resistant Mtb and with a novel mode of action [43]. Compound **47** of the thiazole-hydrazine group has activity against MAH, Mab, and Mtb. Interestingly, pyridine- and thiazol-hydrazides have been shown to display anti-inflammatory and antimicrobial activity [44,45]. In silico docking studies identified an anti-inflammatory mechanism of hydrazine thiazole hybrids through inhibition of the host cyclooxygenase (COX), which is involved in the synthesis of prostaglandins [44].

Due to the intrinsic and acquired ability of mycobacteria to resist killing by antibiotics, multidrug regimens are an important approach for the effective clearance of bacteria and recovery from disease. The aminoglycoside class of the antibiotic AMK is currently a leading antibiotic for the treatment of NTM infections, and INH and RIF are used for Mtb infections. A combination of current antibiotics with novel antimicrobials that target bacterial virulence factors may offer a greater chance for new and more effective therapy regimens. To test if some of the intracellularly active compounds displayed greater activity with current antibiotics and had no antagonist effects, compounds **5**, **10**, **31**, **44**, and **47** were examined for bacterial survival time-kinetics in THP-1 macrophages. The results demonstrate that the tested compounds display substantial bacterial clearance in the antibiotic-compound combination group for MAH, Mab, as well as Mtb when compared with the compound or antibiotic control groups alone. The activity of the combination of compounds was additive, and a few showed no effect. Some of these compounds are likely ideal candidates for medicinal chemistry efforts to identify lead candidates.

Furthermore, while here we mainly detail antimicrobial activity of intracellularly active compounds, those compounds that display activity both in vitro as well as in macrophages could also be promising drug candidates because they exhibit activity against antibiotic resistant NTM clinical isolates and some even at a lower MIC than drug-susceptible MAH 104 strain. Additionally, we note that the combination of bacterial infection with some compound treatments (with intra- and extracellular activity) led to increased toxicity in tissue culture (at nontoxic concentrations to THP-1 cells alone). To resolve the question of the relevance of the antibacterial action of these specific compounds during the host infection, comparisons of cell culture cytotoxicity with in vivo toxicity would be ideal and will specify if these compounds could cause systemic toxicity or if it is only limited to the tissue culture environment.

In this study, overall, we identified 17 clusters of thiazole- and pyridine-hydrazides, triazole- and thiophene-carboxamides, pyrimidine, phenyl-urea and thiourea, triazine diamine, benzamide, and others. Some of these compound groups have been identified from the high-throughput screening of large chemical libraries and are already known for antimicrobial activities, mainly against Mtb. For example, the aminothiazole derivatives have been demonstrated to be medicinally relevant and some highly active in vitro and in silico against Mtb [46]. Using the phenotypic hit targets of Mtb and based on the antitubercular pharmacophore centered screen, pyridine carboxamide prodrugs were described to be activated via KatG-dependent metabolic processing and display inhibitory activity against replicating Mtb [21]. The phenotypic screening of a compound library with known anti-Mtb activity established thiophene-carboxamides as prodrugs activated by the EthA monooxygenase [47]. The thiophene-carboxamide derivatives have also been shown to target a cytidine triphosphate (CTP) synthetase PyrG, which is an essential bacterial factor in different physiological states and, subsequently, compounds had inhibitory activity against Mtb in replicating, non-replicating, and intracellular states and at very low MICs [47]. In addition, triazole hybrids with quinolone antibiotics have been tested in a broad range of clinically important Gram-positive and Gram-negative bacteria and in different physiological states, such as planktonic and biofilms, and were revealed to have greater potency than the antibiotic group alone in all strains [48]. Furthermore, based on the physiochemical analysis and molecular docking studies, the benzamide inhibitors have been identified against *α*-subunit of tryptophan synthase (*α*-TRPS) of Mtb, showing bactericidal activity at 6 μg/mL [49]. The study on 4-amine derivatives that were generated via a structure-guided approach demonstrates that this group of compounds target Mtb cytochrome bd oxidase (Cyt-bd), which is a triheme copper-free terminal oxidase in membrane respiratory chains of bacteria [50]. In addition, this group of compounds represents new antibacterial agents that do not affect the energy metabolism of human organs and tissues [51]. The research on the biological activity of β-arabino glycosyl sulfones against *M. bovis* BCG revealed sulfone compounds as potential inhibitors of mycobacterial cell wall biosynthesis [52].

In summary, our cell-based high-throughput screen identified potent compounds across Mtb and NTM organisms that display activity in different physiological states of replicating and non-replicating bacteria. Some compounds, found for the first time in this study, present promising chemical scaffolds as they demonstrate efficacy against NTM in the host intracellular environment and may offer unique modes of action. Therefore, the findings here encourage further investigations to initiate a medicinal chemistry campaign to utilize structure-activity relationship studies and further optimize the biological activity of a less studied class of compounds in mycobacteria such as the triazine diamines, triazole carboxamides, and thiazole hydrazines.

## Figures and Tables

**Figure 1 molecules-27-05824-f001:**
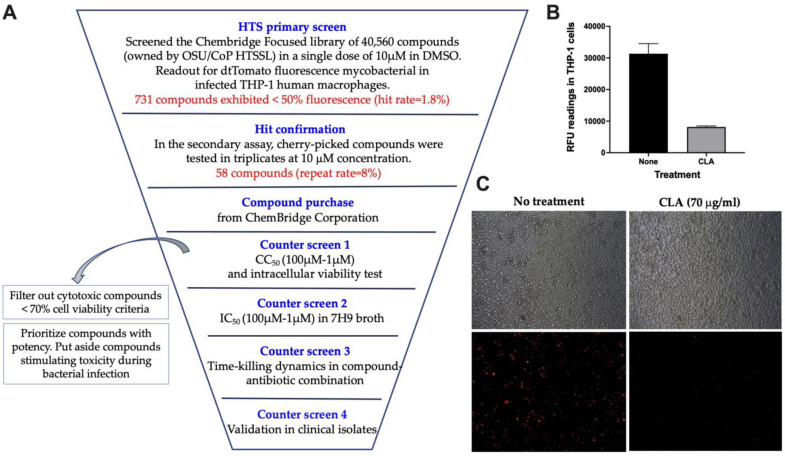
The development of the fluorescent-based HTS assay for identification of compounds with potency in infected macrophages. (**A**) The HTS screen triage. (**B**) Red fluorescence units (RFU) of MAH-T-infected THP-1 macrophages with and without antibiotic treatment at day 7. (**C**) Phase contrast and fluorescent images of MAH-T-infected THP-1 cells with MOI of 40:1 on day 7 post-infection.

**Figure 2 molecules-27-05824-f002:**
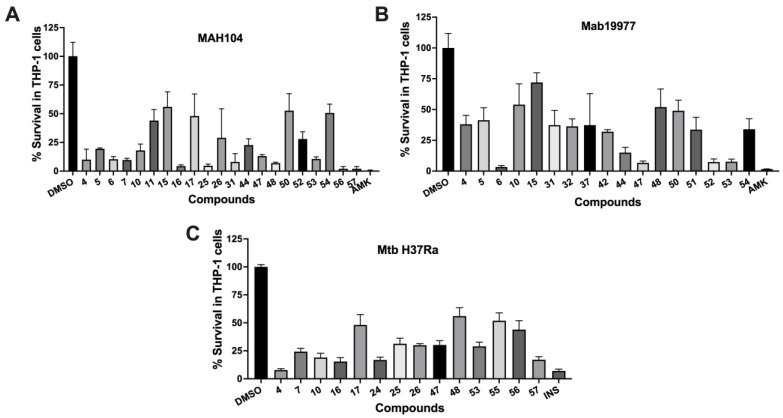
Quantification of bacterial CFUs for assessing intracellular potency of hit compounds against mycobacteria within infected THP-1 macrophages. Compounds at the highest nontoxic concentrations (listed in Table 2) were added to (**A**) MAH, (**B**) Mab, or (**C**) Mtb infected THP-1 cell monolayers at 2 h post-infection. Bacterial CFUs were analyzed by lysing cells with 0.1% Triton X-100 on day 6 for MAH, day 3 for Mab, and day 6 for Mtb and plating seral dilutions on 7H10 agar plates. The percentage of surviving bacteria was calculated by dividing CFU/well for each compound by DMSO growth control which is considered as 100% of bacterial survival. AMK treatment served as a negative control for MAH and Mab growth within macrophages and INH for Mtb. All compounds shown in this figure demonstrated a statistically significant reduction of intracellular bacteria (in *p*-value range of 0.05–0.0001) analyzed with one-way ANOVA multiple comparisons between DMSO control and experimental compound treatment groups.

**Figure 3 molecules-27-05824-f003:**
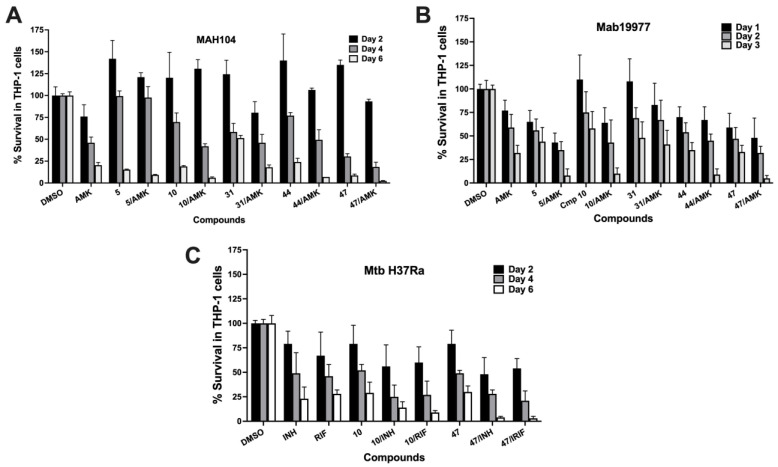
Time-kill dynamics of mycobacteria in THP-1 cells during combination treatment with “intracellular” compounds and antibiotics. (**A**) MAH104 survival rates in macrophages in AMK, compound, and in compound-AMK treatment groups over 6 days of infection. (**B**) Mab19977 survival rates in AMK, compound, and in compound-AMK combination treatment groups over 3 days. (**C**) MtbH37Ra survival rates in INH, RIF, compound, and in compound-antibiotic groups over 6 days of macrophage infection. The intracellular bacterial CFUs were recorded after treatment with antibiotics at the MIC concentration and/or with compounds at the highest nontoxic concentrations to THP-1 cells, and the percentage of surviving bacteria was established by calculating the number of bacteria from the DMSO growth control that received no treatment. Antimicrobials were added to the culture monolayers after 2 h infection. The error bars indicate standard deviations.

**Figure 4 molecules-27-05824-f004:**
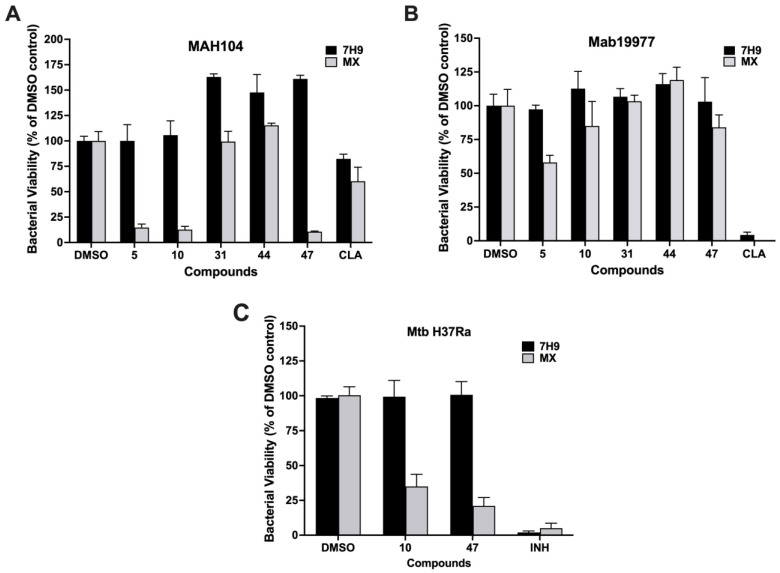
The efficacy of “intracellular” hit compounds against mycobacteria in the metal mix. The activity of compounds was evaluated against (**A**) MAH104, (**B**) Mab19977, and (**C**) MtbH37Ra in the MX to mimicking the metal concentrations and pH of 24 h phagosomes. Selected intracellular compounds were tested at 100 μM concentration, and 1% DMSO was used as a positive control for bacterial growth. Treatment with clarithromycin and isoniazid at bactericidal concentrations served as negative controls for 100% growth inhibition in NTM strains and Mtb, respectively. Bacterial viability was analyzed by utilizing the resazurin reagent read at 530/590 nm excitation/emission filters on a fluorimeter (Tecan). Data represent the means ± standard deviations (SD) of three independent experiments performed in triplicates.

**Table 1 molecules-27-05824-t001:** The list of active compounds identified in this study.

Cluster	Compound	Structure	Name
Thiazole-hydrazine	**24**	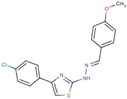	4-(4-chlorophenyl)-N-[(Z)-(4methoxyphenyl)methylideneamino]-1,3-thiazol-2-amine
**25**	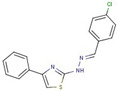	N-[(Z)-(4-chlorophenyl)methylideneamino]-4-phenyl-1,3-thiazol-2-amine
**26**	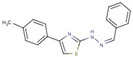	N-[(E)-benzylideneamino]-4-(4-methylphenyl)-1,3-thiazol-2-amine
**47**	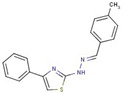	N-[(Z)-(4-methylphenyl)methylideneamino]-4-phenyl-1,3-thiazol-2-amine
Pyridine-hydrazide	**55**	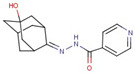	N-[(5-hydroxy-2-adamantylidene)amino]pyridine-4-carboxamide
**56**	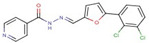	N-[(Z)-[5-(2,3-dichlorophenyl)furan-2-yl]methylideneamino]pyridine-4-carboxamide
**57**	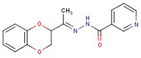	N-[(Z)-1-(2,3-dihydro-1,4-benzodioxin-3-yl)ethylideneamino]pyridine-3-carboxamide
**58**	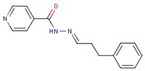	N-[(Z)-3-phenylpropylideneamino]pyridine-4-carboxamide
Triazole-carboxamides	**30**	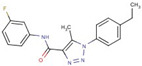	1-(4-ethylphenyl)-N-(3-fluorophenyl)-5-methyltriazole-4-carboxamide
**31**	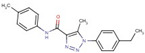	1-(4-ethylphenyl)-5-methyl-N-(4-methylphenyl)triazole-4-carboxamide
**35**	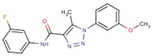	N-(3-fluorophenyl)-1-(3-methoxyphenyl)-5-methyltriazole-4-carboxamide
**37**	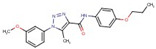	1-(3-methoxyphenyl)-5-methyl-N-(4-propoxyphenyl)triazole-4-carboxamide
Pyrimidine	**19**	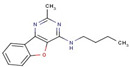	N-butyl-2-methyl-[1]benzofuro[3,2-d]pyrimidin-4-amine
**36**	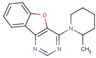	4-(2-methylpiperidin-1-yl)-[1]benzofuro[3,2-d]pyrimidine
**50**	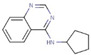	N-cyclopentylquinazolin-4-amine
Phenyl-urea	**44**	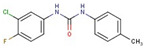	1-(3-chloro-4-fluorophenyl)-3-(4-methylphenyl)urea
**46**	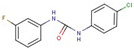	1-(4-chlorophenyl)-3-(3-fluorophenyl)urea
**54**	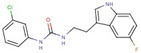	1-(3-chlorophenyl)-3-[2-(5-fluoro-1H-indol-3-yl)ethyl]urea
Triazine diamine	**5**	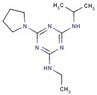	4-N-ethyl-2-N-propan-2-yl-6-pyrrolidin-1-yl-1,3,5-triazine-2,4-diamine
**10**	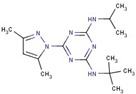	2-N-tert-butyl-6-(3,5-dimethylpyrazol-1-yl)-4-N-propan-2-yl-1,3,5-triazine-2,4-diamine
Benzamide	**16**	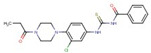	N-[[3-chloro-4-(4-propanoylpiperazin-1-yl)phenyl]carbamothioyl]benzamide
**18**	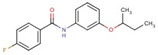	N-(3-butan-2-yloxyphenyl)-4-fluorobenzamide
Thiophene-carboxamide	**11**	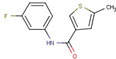	N-(3-fluorophenyl)-5-methylthiophene-3-carboxamide
**51**	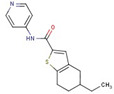	5-ethyl-N-pyridin-4-yl-4,5,6,7-tetrahydro-1-benzothiophene-2-carboxamide
Alcohol with piperazine	**17**	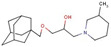	1-(1-adamantylmethoxy)-3-(3-methylpiperidin-1-yl)propan-2-ol
**42**	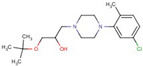	1-tert-butoxy-3-[4-(5-chloro-2-methylphenyl)-1-piperazinyl]-2-propanol hydrochloride
Thiourea	**48**	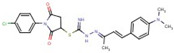	[1-(4-chlorophenyl)-2,5-dioxopyrrolidin-3-yl] N’-[(Z)-[(E)-4[4(dimethylamino)phenyl]but-3-en-2-ylidene]amino] carbamimidothioate
**53**	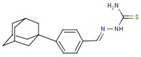	[(Z)-[4-(1-adamantyl)phenyl]methylideneamino]thiourea
Sulfone	**15**	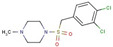	1-[(3,4-dichlorophenyl)methylsulfonyl]-4-methylpiperazine
**49**	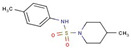	4-methyl-N-(4-methylphenyl)piperidine-1-sulfonamide
Pyridine-carboxamides	**4**	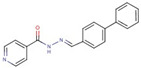	N’-(4-biphenylylmethylene)isonicotinohydrazide
Carboxylic acid	**6**	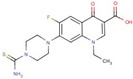	7-(4carbamothioylpiperazin-1-yl)-1-ethyl-6-fluoro-4-oxoquinoline-3-carboxylic acid
Aminothiazole	**7**	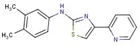	N-(3,4-dimethylphenyl)-4-pyridin-2-yl-1,3-thiazol-2-amine
Acetamide	**32**	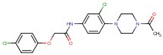	N-[4-(4-acetylpiperazin-1-yl)-3-chlorophenyl]-2-(4-chlorophenoxy) acetamide
Imidazole	**40**	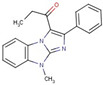	1-(4-methyl-2-phenylimidazo[1,2-a]benzimidazol-1-yl)propan-1-one
Pyridine-thioether	**52**	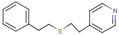	4-[2-(2-phenylethylsulfanyl)ethyl]pyridine

**Table 2 molecules-27-05824-t002:** Activity of hit compounds in THP-1 macrophages and in vitro.

Compound	THP-1 Cytotoxicity [μM]	Intracellular Killing in THP-1 Cells			MIC_50_ [μM]		
		MAH104	Mab19977	MtbH37Ra	MAH104	Mab19977	MtbH37Ra
**4**	32	Yes	Yes	Yes	10	-	3
**5**	32	Yes	Yes	-	-	-	-
**6**	-	Yes	Yes	-	3	32	10
**7**	10	Yes	-	Yes	10	10	10
**10**	32	Yes	Yes	Yes	-	-	-
**11**	32	Yes	-	-	-	32	-
**15**	-	Yes	Yes	-	-	-	-
**16**	10	Yes	-	Yes	32	-	-
**17**	32	Yes	-	Yes	100	-	10
**18**	32	*	-	-	100	-	-
**19**	-	-	-	-	100	100	10
**24**	-	-	-	Yes	-	-	-
**25**	-	Yes	-	Yes	32	-	10
**26**	32	Yes	-	Yes	-	-	10
**30**	-	-	-	-	-	-	10
**31**	-	Yes	Yes	-	-	-	-
**32**	32	-	Yes	-	-	-	-
**35**	32	*	-	-	-	100	-
**36**	32	*	*	-	100	100	-
**37**	32	-	Yes	-	100	-	-
**40**	32	-	-	-	-	-	10
**42**	-	-	Yes	-	-	-	10
**44**	-	Yes	Yes	-	-	-	-
**46**	32	*	*	-	10	32	-
**47**	-	Yes	Yes	Yes	-	-	-
**48**	32	Yes	Yes	Yes	10	-	10
**49**	32	-	-	-	-	-	10
**50**	-	Yes	Yes	-	-	-	-
**51**	32	-	Yes	-	-	-	10
**52**	-	Yes	Yes	-	10	-	-
**53**	32	Yes	Yes	Yes	-	-	10
**54**	10	Yes	Yes	-	32	-	10
**55**	-	*	-	Yes	32	-	10
**56**	32	Yes	-	Yes	100	10	-
**57**	32	Yes	-	Yes	32	-	-
**58**	-	*	-	-	100	-	-

Note: for active compounds that are marked with asterisk, bacterial infection increases the toxicity to THP-1 cells at nontoxic concentrations.

**Table 3 molecules-27-05824-t003:** Effect of compounds against *M. avium* and *M. abscessus* clinical isolates in vitro.

Compound	MIC_50_ [μM]							
	MAH104	NJH 0133	MAH B	MAH C	Mab19977	DNA 01627	NR49093 Strain DJO44274	NR44273 Strain 4529
**4**	10	10	100	10	-	-	-	-
**6**	3	3	-	3	32	32	32	32
**7**	10	10	-	-	10	10	10	10
**11**	-	-	-	-	32	32	32	32
**16**	32	32	100	100	-	-	-	100
**17**	100	100	100	-	-	-	-	-
**18**	100	100	100	100	-	-	-	100
**19**	100	100	100	100	100	100	100	100
**25**	32	-	-	-	-	-	-	-
**35**	-	-	-	-	100	100	32	32
**36**	100	100	-	100	100	100	32	32
**37**	100	100	-	100	-	-	-	-
**46**	10	10	-	10	32	32	10	32
**48**	10	10	32	32	-	-	-	-
**52**	10	-	-	-	-	-	-	-
**54**	32	100	-	-	-	-	-	-
**55**	32	10	10	10	-	-	-	-
**56**	100	10	10	10	10	10	32	32
**57**	32	10	10	100	-	-	-	-
**58**	100	10	10	100	-	-	-	-

## Data Availability

Not applicable.

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
