# Peer review of "Identification of Small Molecule Inhibitors against Mycobacteria in Activated Macrophages"

_molecules, 2022, doi:10.3390/molecules27185824_

Round 1

Reviewer 1 Report

In figure 1: remove the red underline of chemBridge snapped picture

Author Response

In figure 1: remove the red underline of chemBridge snapped picture

A: We removed the red underline from the Figure 1. Thank you.

Reviewer 2 Report

Figure 1, please let A, B, and C on the figure.

Line 49 replace "agent" by "species"

Replace the words "cherry picked" with "randomized"

Author Response

Figure 1, please let A, B, and C on the figure.

A: Letters A, B, and C have been added to the Figure 1.

Line 49 replace "agent" by "species"

A: “Agent” has been replaced by “species” in line 49. 

Replace the words "cherry picked" with "randomized"

A: This sentence was changed with the following:

Lines 164-165: The hit compounds were selected from the library and were retested in the secondary assay in three technical replicates as described above.

Reviewer 3 Report

Voorde et al. developed and performed a HTS screen to identify novel inhibitors against mycobacteria. They identified several bioactive compounds that target different physiological states of replicating and non-replicating bacteria. Co-treatment of these new inhibitors with currently used antibiotics caused greater bacterial clearance compared with compound/antibiotics alone. Some of the inhibitors discovered in this study have novel chemical scaffolds that could be potentially interesting for further potency optimization in medicinal chemistry campaigns. Overall, this is a well-written manuscript with good study design and good discussion into the putative protein targets that their new agents could possibly inhibit. Some minor issues need to be addressed before the manuscript can be accepted for publication.

1. Because a large number of compounds (40,560) were screened, please provide more information on how robust the screen was? Was the Z’ factor for the assay determined? Please comment on the inter-day reproducibility, if any?

2. Line 53-54: Please correct this sentence. Clarithromycin and azithromycin are macrolides. Amikacin is an aminoglycoside. Cefoxitin belongs to the cephalosporin class.

3. Please amend all mentions of “Triton-X-100” to “Triton X-100”.

4. Line 221: Should be “fluorescence-based”.

5. Line 426: Please edit for clarity.

6. Line 464: Should be “drug candidates”.

7. Line 477: Should be “aminothiazole”. Also, correct the "aminothiazol" typological error in Table 1.

Author Response

  1. Because a large number of compounds (40,560) were screened, please provide more information on how robust the screen was? Was the Z’ factor for the assay determined? Please comment on the inter-day reproducibility, if any?

A: New lines 231-232. The Z’-score was determined to be 0.55, which is considered a good quality assay[1]. 

  1. Lines 53-54: Please correct this sentence. Clarithromycin and azithromycin are macrolides. Amikacin is an aminoglycoside. Cefoxitin belongs to the cephalosporin class.

A: New lines 52-54. Most often two macrolides, such as clarithromycin and azithromycin, along with an aminoglycoside, commonly amikacin, and the cephalosporin cefoxitin are used for the initial phase of treatment.

  1. Please amend all mentions of “Triton-X-100” to “Triton X-100”.

A: This change has been made.  Thank you.

  1. Line 221: Should be “fluorescence-based”.

A: This change has been made.

  1. Line 426: Please edit for clarity.

A: Lines 427-430 have been edited for clarity as follows: Compounds 5,10, 31, 44 and 47 demonstrated bactericidal activity against MAH in MX, while only three compounds (5, 10 and 47) inhibited Mab growth in MX. In addition, compounds 10 and 47 displayed inhibitory activity against Mtb in MX media.

  1. Line 464: Should be “drug candidates”.

New line 467: This change has been made.

  1. Line 480: Should be “aminothiazole”. Also, correct the "aminothiazol" typological error in Table 1.

A: This change has been made in the Table 1 and in the text line 479.  Thank you.